# Integrating Protein Quality and Quantity with Environmental Impacts in Life Cycle Assessment

**Andrew Berardy [1],\***, **Carol S. Johnston [2]**, **Alexandra Plukis [3]**, **Maricarmen Vizcaino [2]** and **Christopher Wharton [2],\***

1  Swette Center for Sustainable Food Systems, Arizona State University, 800 Cady Mall, Tempe, AZ 85281, USA
2  College of Health Solutions, Arizona State University, 550 N 3rd St, Phoenix, AZ 85004, USA; CAROL.JOHNSTON@asu.edu (C.S.J.); mvizcain@asu.edu (M.V.)
3  School of Computing, Informatics, and Decision Systems Engineering, Arizona State University, 699 S Mill Ave, Tempe, AZ 85281, USA; aplukis@asu.edu
\*  Correspondence: aberardy@asu.edu (A.B.); Christopher.Wharton@asu.edu (C.W.)

**Abstract:** Life cycle assessment (LCA) evaluates environmental impacts of a product from material extraction through disposal. Applications of LCA in evaluating diets and foods indicate that plant-based foods have lower environmental impacts than animal-based foods, whether on the basis of total weight or weight of the protein content. However, LCA comparisons do not differentiate the true biological value of protein bioavailability. This paper presents a methodology to incorporate protein quality and quantity using the digestible indispensable amino acid score (DIAAS) when making comparisons using LCA data. The methodology also incorporates the Food and Drug Administration's (FDA) reference amounts customarily consumed (RACCs) to best represent actual consumption patterns. Integration of these measures into LCA provides a mechanism to identify foods that offer balance between the true value of their protein and environmental impacts. To demonstrate, this approach is applied to LCA data regarding common protein foods' global warming potential (GWP). The end result is a ratio-based score representing the biological value of protein on a GWP basis. Principal findings show that protein powders provide the best efficiency while cheeses, grains, and beef are the least efficient. This study demonstrates a new way to evaluate foods in terms of nutrition and sustainability.

**Keywords:** life cycle assessment; protein; diet; sustainability; environmental impacts; plant-based diets

## 1. Introduction

As awareness of the environmental consequences of food choices grows, consumers' demand for plant-based alternatives to meat and other animal-based foods has increased as well, in part due to a perception of plant foods' lesser impact on the environment [1,2]. The Nielsen Company noted in 2017 that 23% of North American consumers wanted more plant-based proteins available and 39% of Americans were actively trying to incorporate more plant-based foods into their diets to improve health and nutrition, eat "clean," save money, and protect the environment [2]. In 2018, another Nielsen study found 20% growth in sales of plant-based foods, up from 8% in 2017 and compared to 2% overall growth, which may be due to the large variety of high-quality plant-based foods now available to consumers [1].

Given this market trend, two primary issues arise in relation to non-animal-based protein foods, specifically their contribution to the quality of the diet and their implications for the environment. Separate methodologies currently exist to evaluate data regarding each. These include life cycle

assessment (LCA), a common choice for assessing the environmental impacts of food, and the digestible indispensable amino acid score (DIAAS), an advanced evaluation of how amino acids are digested and assimilated by the body, providing a score related to "protein quality".

Environmental impacts can be evaluated through LCA by quantifying resource usage, pollutant emissions, and other potential impacts of consumer products [3]. These measures represent flows between the biosphere and the technosphere associated with the creation, consumption, and disposal of one or more products, including food [4]. Although LCA was originally developed for application to industrial production systems, scholars applied its principles to agricultural production systems over two decades ago [5]. Nearly all LCAs of food include global warming potential (GWP) as an environmental impact of concern, while other common metrics considered include energy use, water use, and land use [4,6]. As an example, dietary LCA assessments of standardized plant-based diets (often constructed to represent one day's worth of dietary intake using such a pattern) had lower environmental impacts across most environmental impact categories than diets including animal products [7–11].

LCA requires a functional unit that captures the obligatory properties of a product for use as a basis for comparison across products on environmental impact [12]. Obligatory properties are those properties that a product must have in order to be considered as an appropriate alternative when compared with other products [12]. However, functional units for LCAs of food are difficult to define due to variation in perceived obligatory properties. For example, a typical approach for food is to use a mass-based functional unit, such as tons or kilograms of product. In many cases, a weight-based comparison between two different foods is inadequate because it does not reflect the amount that a person would typically consume, nor does it capture the reasons a particular food would be eaten. For instance, comparing beef against broccoli on a per-kilogram basis would be inappropriate, given that both are usually eaten in different amounts and for different reasons.

For high protein foods, a protein-based functional unit is logical, but insufficient to represent the actual bioavailability of amino acids, which is the assumed purpose of protein intake. This issue can possibly be corrected with the consideration of other data. The calculation of DIAAS is the latest advance in comparing the protein qualities of foods, and this assessment has the potential to shift perceptions regarding what foods are good protein sources. DIAAS is a scoring system that compares the true ileal digestibility of the indispensable amino acids in a protein food source to that of a reference protein. This scoring system replaced the protein digestibility corrected amino acid score (PDCAAS), which was the reference standard of the United Nations for scoring protein quality from 1989 to 2011 [13]. PDCAAS used the digestibility of the protein for scoring (e.g., the digestibility of nitrogen), which is a less precise indicator of true amino acid assimilation in comparison to the DIAAS method [14]. To calculate the DIAAS percentage for any food, the milligrams of digestible dietary indispensable amino acids in one gram of the dietary protein is divided by the milligrams of the amino acids in one gram of the reference protein, then multiplied by 100 [13]. The DIAAS scoring system allows foods to have an amino acid quality score over 100%, while the PDCAAS method truncated scores at 100% [13]. As such, DIAAS scores provide the most accurate ranking of protein foods since the score depicts the actual, unabridged biological value of the protein's amino acids. DIAAS can be used in the analysis of meal plans to determine the adequacy of protein sources to meet recommended intakes (0.8g/kg of body weight/day), which is particularly relevant for individuals following vegetarian and vegan diets [14,15].

Yet another issue that arises in comparing foods using LCA is how best to reflect results in terms of real-world serving sizes. Although LCA and DIAAS provide useful tools to compare foods on environmental impact and protein quality, it is less clear how meaningful these comparisons are in everyday settings given that customary serving sizes of foods can vary widely. This problem can potentially be solved using reference amounts customarily consumed (RACCs). In accordance with the Nutrition Labeling and Education Act and the Federal Food, Drug, and Cosmetic Act, the Food and Drug Administration's final rule to establish RACCs provides a reference for identifying actual

typical consumption amounts balanced with recommended portions [16–18]. RACCs defined by the FDA help manufacturers determine serving sizes for the Nutrition Facts label on foods based on a balance of the typical amount of food people eat and a recommended portion. The FDA continues to update RACCs to provide the most accurate and current information for serving sizes and improve the Nutrition Facts panel [19]. Utilizing RACCs as part of LCA assessments therefore can provide a clearer understanding of the environmental impact related to actual consumption of foods.

Previous LCA research has yet to provide a straightforward methodology for meaningful evaluation of foods in the realms of health and environmental impact simultaneously. Previous LCA work has expressed results using functional units based on a unit of mass (e.g., kilogram of a given food) or a unit of a certain nutrient (e.g., kilograms of protein) and expressed impacts in terms of estimated equivalent flows into the biosphere (e.g., kilograms of carbon dioxide) or aggregated multiple characteristics into a single score (e.g., Ecoindicator 99) [7,20]. Some papers also attempted to integrate multiple dimensions of sustainability in one assessment, such as combining LCA with cost-benefit analysis and social impacts as measured by quality adjusted life years (QALYs) [12]. No work yet exists employing an algorithm that adequately assesses environmental impact of protein-based foods in the context of protein quality. Further, few LCA analyses have developed methods to express results in ways that reflect consumer food choices and consumption patterns.

This paper presents a methodology to incorporate protein quality and quantity using DIAAS when making comparisons based on LCA results. Evaluations were further integrated with RACCs to better align estimates of environmental impacts with an understanding of how much is actually consumed and what amino acids are actually delivered in real-world servings of foods. Serving sizes provided by RACCs were specifically used as a basis for calculating a weighted protein score with associated environmental impacts. The intent was to align serving sizes with what is customarily consumed, providing more holistic information that could potentially be utilized in decision-making related to a healthy and sustainable diet simultaneously.

## 2. Materials and Methods

The metric chosen for this study to represent environmental impact of foods was carbon dioxide ($CO_2$) equivalents. This was done for three reasons: first, $CO_2$ equivalents are a measure of primary concern included in nearly all LCA studies to express greenhouse gas emissions associated with global warming potential throughout a product's lifecycle [4,6,20,21]. Second, global warming is used frequently as a justification when calling for dietary change [22,23]. Third, while other LCA studies include metrics other than $CO_2$ equivalents such as land use, water use, eutrophication, and acidification, such metrics are less consistent in terms of methodology for their estimation and in their inclusion across articles [24–26].

Using data from the United States Department of Agriculture's (USDA) Economic Research Service (ERS) Food Availability (Per Capita) Data System (FADS), we identified the most common protein sources for Americans [27]. Due to the importance of protein across dietary choices, we also included the most common sources of animal, dairy, and plant protein intake for adults in the US based on analysis of the National Health and Nutrition Examination Survey (NHANES) data [28]. We additionally included protein sources that are gaining popularity as potentially more sustainable alternative sources of proteins, including plant- and insect-based proteins [29,30]. Mixed dishes and foods with many ingredients were also excluded due to our focus on protein and lack of available LCA data. Lack of data regarding global warming potential or DIAAS values for certain foods resulted in a narrowed list of protein sources. Some of the protein sources that were excluded due to lack of data availability were mealworms, flies, and various beans, nuts, and breads. Finally, liquid protein sources, such as milk were not included due to concerns with comparing solid and liquid foods, such as liquid foods having a much higher weight per serving size. Figure 1 illustrates the process for determining the final list of foods included.

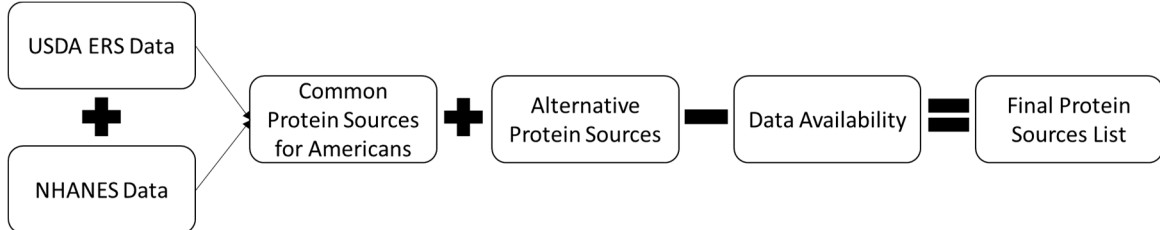

**Figure 1.** The list of protein sources considered was based on common and alternative protein sources, but constrained by data availability.

Data from the USDA ERS FADS regarding nutrient availability indicated that food groups contributing the highest percent to total protein availability include meat, poultry, and fish, grain products, dairy products, legumes, nuts, and soy, vegetables, and eggs [27].

Figure 2 displays the percentage of available protein across a variety of food groups from the USDA ERS FADS data. Among animal sources of protein, beef, pork, and chicken consistently have the highest availability for consumers in the US, with chicken overtaking beef as the most consumed meat in 2015, while fish and shellfish remain steady over time below the other three [31]. We additionally included foods providing 2% or more to total animal protein, dairy protein, or plant protein intake based on analysis of NHANES data by Pasiakos et al. [28]. Alternative protein sources including plant-based foods and insects are receiving increased attention as potentially more sustainable protein sources [29,30,32]. Based on these trends, we included foods representing each major protein source category as well as some niche but growing categories that represent alternative and potentially more sustainable protein sources, all subject to data availability.

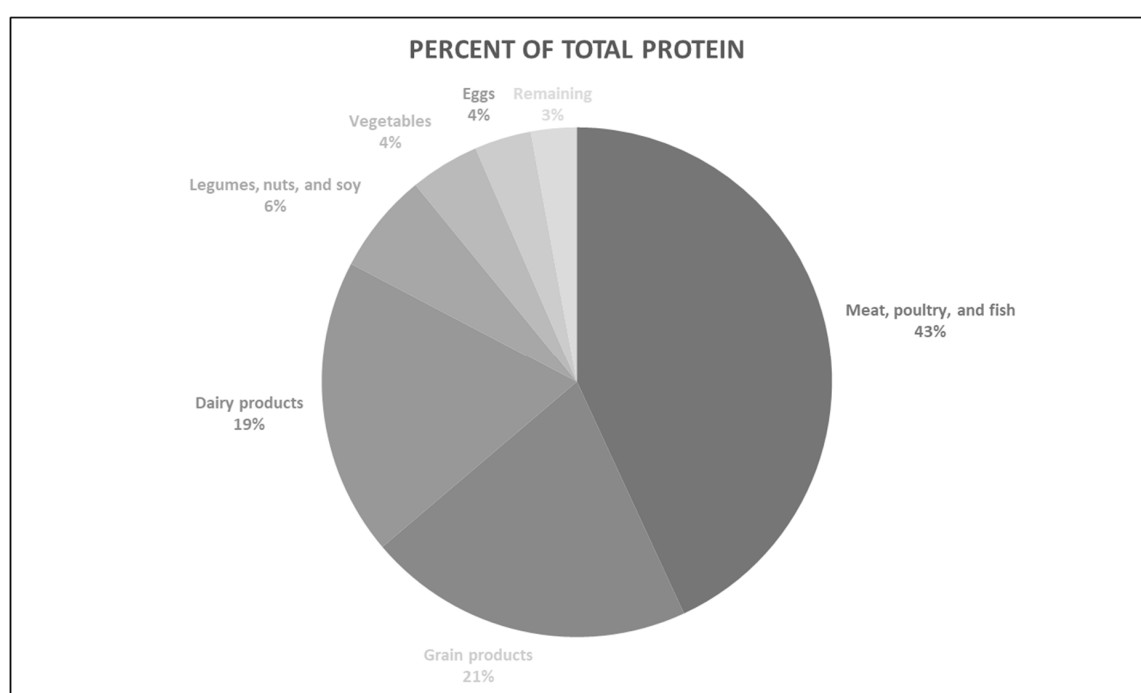

**Figure 2.** Protein availability across food groups. Animal flesh, grains, and dairy provide most protein available for US consumers.

Amino acid profiles of foods were obtained using Food Processor Nutrition Analysis software version 11.4, a searchable nutritional database that provides information on micro and macronutrients present in raw, cooked, and restaurant-prepared foods (ESHA Research, Salem, OR, USA). We utilized protein-specific data from individual food searches, prioritizing information provided by the USDA. We calculated a weighted protein score (PRO) as shown in Formula (1) based on the foods' protein

content and DIAAS. PRO equals the product of DIAAS (D) as a percentage, servings size in grams based on RACCs (R), and grams of protein weight in 100 grams of a given food (P), divided by 100.

$$PRO = \frac{\frac{D}{100} \times R \times P}{100} \tag{1}$$

To represent GWP, $CO_2$ emissions data were compiled from dozens of LCA sources including peer-reviewed published literature. Similar to previous reviews of food LCA, data availability restraints required the use of some LCA studies which included different assumptions regarding co-product allocation methodology and system boundaries, different geographic areas, and other factors that could result in disparities in estimates of environmental impacts [4,21,33,34]. While most studies report carbon footprints as point values, there can be significant uncertainty and variability surrounding such point values [35]. Variability can be a result of differences in farm features, soil type, and management practices [36]. Reviewing the parameters of LCA data sources included indicated that the most common attributes were a system boundary of cradle to gate, economic-based allocation, attributional LCA, and a European region. However, excluding studies with different parameters than these would restrict the analysis to only include beef, chicken, eggs, milk, and pork, as was the case in a 2010 review of livestock LCA [21]. All LCA data sources used for our analysis were subject to peer review and used system boundaries at least from cradle to gate. In addition, we only used studies that expressed results with at least a mass-based functional unit and global warming potential in a mass-based $CO_2$ equivalent. The majority of references were from Organisation for Economic Co-operation and Development member countries, although data availability necessitated the inclusion of some non-members (e.g., Thailand for grasshoppers and Iran for chickpeas, lentils and peanuts) [37–40]. Finally, where multiple LCA data sources for individual foods were available that met our requirements, we calculated mean and standard deviation for the food, which are shown in Figure 3. Items with an asterisk had fewer than three entries and therefore are based on the only reported value or an average of two reported values and do not include a standard deviation. Further details regarding our LCA data sources and calculations are available in the Supplementary Materials.

$$GWPRO = \frac{GWP}{PRO} \tag{2}$$

Finally, we calculated a protein-mediated modified global warming potential ratio (GWPRO) as shown in Formula (2) to capture protein delivery across foods in our comparison through a single score metric to allow straightforward ranking of alternatives that includes consideration of both GWP and PRO.

## 3. Results

The GWP associated with 100 grams of a given food has a substantial range as shown in Figure 3, which represents GWP of foods on a mass basis using data from previously published LCA work. Where sufficient data was available, a mean and standard deviation was calculated. Foods with only two data points are presented as an average of those two while those with only one data point are reported as that value, and both are marked with an asterisk and do not include standard deviation.

It is clear from Figure 3 that GWP on a mass basis is highest by far for beef, and that in general, animal-based foods, including dairy products, are higher in GWP than plant-based foods. However, this representation of data does not capture the protein content in terms of quality and quantity and therefore is insufficient to make a determination for what foods to eat in order to strike the best balance between meeting protein needs and minimizing environmental impact. To better assess that balance, we use Formula 2 to capture the PRO score and GWP associated with a typical serving according to RACCs in a single score format. Figure 4 displays the resulting GWPRO ratios for the same foods as shown in Figure 3.

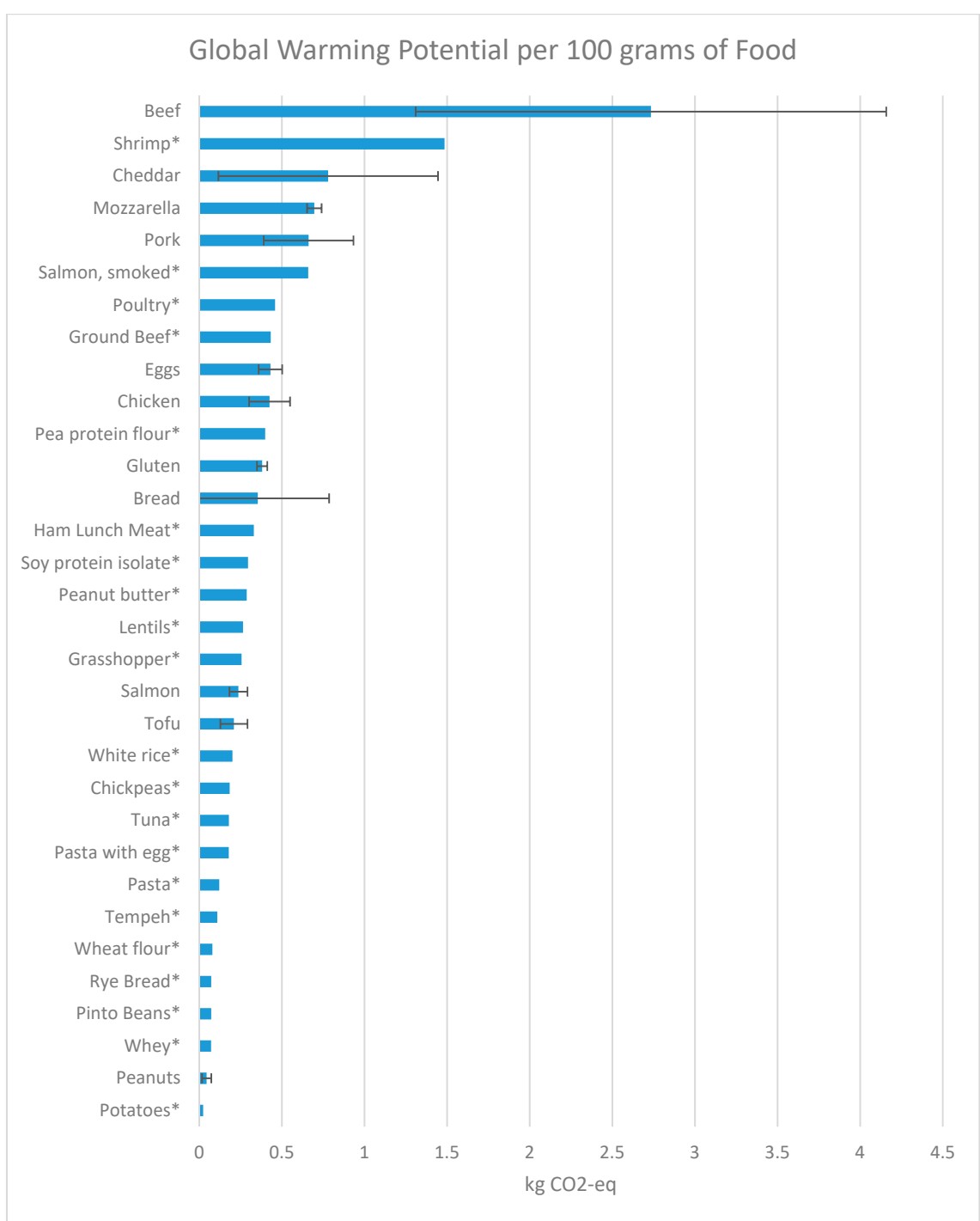

**Figure 3.** Global Warming Potential (GWP) in kilograms of $CO_2$ equivalents associated with 100 grams of each food. Values are the mean of three or more entries if there was sufficient data available. Error bars indicate standard deviation away from the mean across multiple LCA data points when three or more were available. Items marked with *asterisk had fewer than three data points and therefore do not include standard deviation and are either the average of two values or just reporting of one value, depending on data availability.

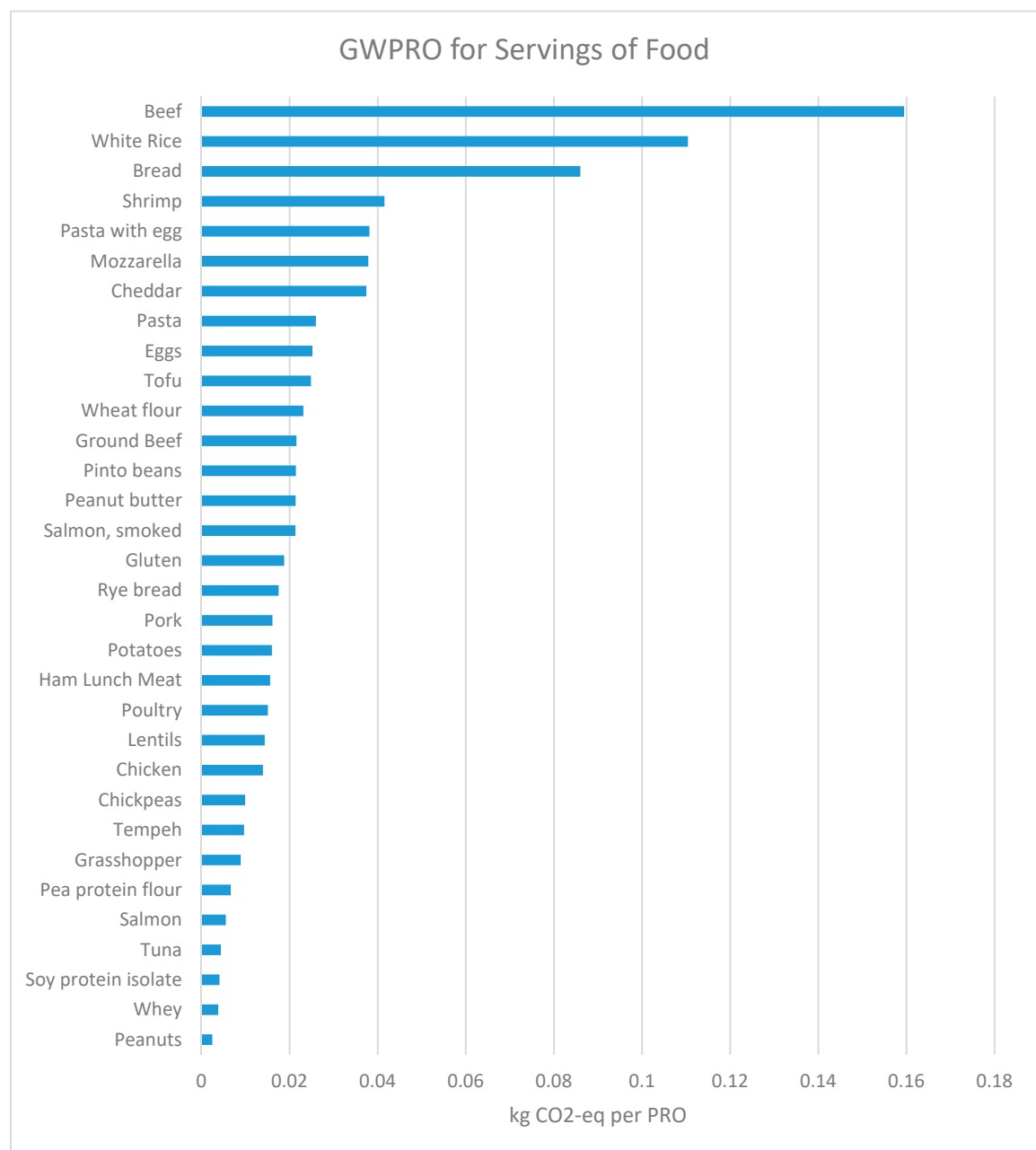

**Figure 4.** GWPRO (Global Warming Potential Ratio) is GWP in kg CO2-eq per serving divided by PRO per serving to capture the protein content and quality in the environmental comparison of protein sources. Standard deviation is not reported here as values are the result of a ratio with PRO. However, data used for the GWP component are the same as that used for Figure 3.

As seen in Figure 4, the ranking of foods changes when protein content and quality is taken into consideration. For example, spaghetti, despite having the second lowest GWP on a mass basis, has a fairly low EPE ratio and is outperformed by foods with much higher GWP that also have higher PRO scores. Using this algorithm, various protein powders are shown to be some of the most efficient foods considered, and plant-based foods have a broader spread across the list than in Figure 3 due to their wide variation in protein content and quality. It is also important to note that these impacts are partially a reflection of their disparate serving sizes, which result in lower impacts for lower servings when compared to a uniform weight across foods. However, it is this specific distinction that should help improve understanding of actual consumption decisions by providing a comparison based on

amounts that are customarily consumed. The method demonstrated here is a potential way to include consideration of protein quality and serving size along with environmental performance in LCA. Consumers should understand that these are estimates meant for quick and efficient comparisons that are informative but not comprehensive and are subject to change. Environmental impacts of a food are influenced by many factors that vary across regions and over time, and many factors influence the nutritional quality and ultimate health impacts of a given food.

## 4. Discussion

Previous studies that examined the environmental and nutritional impacts of protein sources included a variety of animal and plant-based foods, but did not specify inclusion and exclusion criteria except in some cases to match dietary intake within data constraints [20,34,41–43]. Our study contributes to the advancement of LCA-based food comparisons by expanding and refining the methodology to account for variation in amounts of food typically consumed and differences in protein quality and quantity.

The use of serving size as a basis for comparison of environmental impacts means that the results can be made useful for consumers, but they may not be as readily applicable to large-scale estimates as strictly uniform weight-based comparisons normally are. The conversion between a mass basis and a serving size basis is simple when the serving size is known, but initial choice must take into account the intended audience. These data, for example, could be used to establish a consumer-facing scoring or rating system, allowing comparisons of possible food choices serving similar roles in the diet and at similar serving sizes (e.g., ground beef vs. tuna vs. tofu) along the lines of dietary impact related to protein quality and environmental impact at the same time.

## 5. Conclusions

As more consumers seek to change their eating habits to address concerns related to environmental impact, it is important to develop evidence-based tools that allow for consideration of, and distinguishing between, food choices across multiple relevant concerns. In the case of high-protein foods, often the most resource-intensive foods in the diet, integration of serving size and protein quality and quantity can enhance an LCA study by providing a more complete understanding of alternatives. This is especially important when comparing plant-based and animal-based foods, as protein is the primary nutrient of concern in such scenarios. The DIAAS provides the most accurate currently available information regarding protein quality through amino acid digestibility, and when combined with protein quantity through weight of protein in a food, a weighted protein score provides a method for quick and comprehensive comparison across foods. Integrating these concepts into LCA is demonstrated here to effectively differentiate between foods that have varying levels of protein content and environmental impacts and identify the best and worst performing foods for both metrics at once. The best performing foods include peanuts, whey, and soy protein isolate, while the worst performing foods include beef, white rice, and bread. Data constraints prevented a comprehensive analysis of alternatives, but such challenges are common in studies aggregating data from multiple LCA sources. Although multiple LCA data sources were used where available, regional and temporal variability as well as other factors including management practices and transportation distances all influence the environmental impact associated with any given food. In addition, food consumption patterns do not always match RACCs, recommended servings, or other nutritional guidance, so actual food consumption will vary with the potential for dramatic impacts on the environmental consequences of food choices. Further work to improve this methodology could include data related to macronutrients other than protein when they may also contribute to the obligatory property of a food and thus serve as a functional unit, the inclusion of environmental impacts besides greenhouse gasses, and the inclusion of economic considerations such as affordability and accessibility.

**Supplementary Materials:** The following are available online at http://www.mdpi.com/2071-1050/11/10/2747/s1.

**Author Contributions:** Conceptualization, A.B. and C.W.; methodology, A.B., C.W., C.J. and M.V..; software, A.B., M.V. and A.P.; validation, A.B. and M.V.; formal analysis, A.B.; investigation, A.B., M.V., and A.P.; resources, C.W. and C.J.; data curation, A.B.; writing—original draft, A.B.; writing—review and editing, A.B., C.W., M.V., and C.J.; visualization, A.B., C.W., M.V., and C.J.; supervision, C.W.; project administration, C.W.

**Funding:** This research received no external funding.

**Conflicts of Interest:** The authors declare no conflicts of interest.

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
