# Peer review of "Integrating Protein Quality and Quantity with Environmental Impacts in Life Cycle Assessment"

_sustainability, doi:10.3390/su11102747_

Round 1
Reviewer 1 Report
The authors deal with a controversial issue, which is how to make a fair comparison of alternative protein sources. I find that the proposed weighted protein score (WPS) can be very useful, since the primary goal of food is nutrition, and related to this, the content in essential of amino acids is crucial when comparing protein sources. However, there are some issues that should be improved to increase the quality of the article.
- The first one is related to the data sources used for the CFP. When explaining the LCA data sources (line 170), I miss more details. For instance, which system boundaries were considered when searching the LCA data? Which FU? From which geographic area (origin)? I recommend the authors to include all this details in a table. The table from the supplementary material in Clune et al. (2017) (reference [31]) can be taken as an example.
- Also related with the GWP is the variability of the CFP data, since depending on factors such as the management practices, geographic area were the food is produced etc., the results of the GWP can vary. From my point of view, including this variability, e.g. calculating confidence intervals, or at least the standard deviation, would increase the reliability of the results shown in Figures 3 and 4 and would reinforce the quality of the paper. The following papers could be used as a guide:
Henriksson PJ, Heijungs R, Dao HM, Phan LT, de Snoo GR, Guinée JB. (2015) Product carbon footprints and their uncertainties in comparative decision contexts. PLoS One 10(3):e0121221
Ribal, J., Estruch, V., Clemente, G., Fenollosa, M. L., & Sanjuán, N. (2019). Assessing variability in carbon footprint throughout the food supply chain: a case study of Valencian oranges. The International Journal of Life Cycle Assessment, 1-18.
- As to the serving, I agree with the authors that it reflects the amount of food consumed nowadays in the USA, but as the authors also state (line 213) they are subject to change, therefore they are also variable. In addition, serving sizes do not always agree with nutritional recommendations. Along these lines, and taking into account that according to the last USDA recommendations the, food pyramid has become a dish, it would be interesting to show some examples comparing two (or more) dishes, e.g. vegetarian vs. omnivorous.
- In the conclusions, I recommend to add some comments about the above issues: variability of the carbon footprint and the serving size.
Minor changes: Line 129. Please, include the reference of the data sources
Author Response
Response to Reviewer 1 Comments
Point 1: The first one is related to the data sources used for the CFP. When explaining the LCA data sources (line 170), I miss more details. For instance, which system boundaries were considered when searching the LCA data? Which FU? From which geographic area (origin)? I recommend the authors to include all this details in a table. The table from the supplementary material in Clune et al. (2017) (reference [31]) can be taken as an example.
Response 1: We updated the text to provide additional details regarding parameters of included LCA studies. Additionally, we include an Excel file with full details for each study, similar to the recommended supplementary material from Clune et al. 2017. We would like to make this available as an Appendix or supplementary material so that readers can review detailed characteristics for each data point. We understand that there is variation in the parameters of LCA studies included in our analysis. It was unclear to us if Reviewer 1 intended for us to ensure such parameters were all similar for included studies or if they only intended for us to include this information as a disclaimer regarding values reported from different items having different assumptions. As noted in the manuscript, previous reviews of food LCA have included differing assumptions regarding allocation and system boundaries, as well as differing geographic areas and other factors. When considering the option of restricting data sources to only those with all identical assumptions and parameters we realized that this would dramatically narrow the scope of the foods considered.
Point 2: Also related with the GWP is the variability of the CFP data, since depending on factors such as the management practices, geographic area were the food is produced etc., the results of the GWP can vary. From my point of view, including this variability, e.g. calculating confidence intervals, or at least the standard deviation, would increase the reliability of the results shown in Figures 3 and 4 and would reinforce the quality of the paper. The following papers could be used as a guide:
Henriksson PJ, Heijungs R, Dao HM, Phan LT, de Snoo GR, Guinée JB. (2015) Product carbon footprints and their uncertainties in comparative decision contexts. PLoS One 10(3):e0121221
Ribal, J., Estruch, V., Clemente, G., Fenollosa, M. L., & Sanjuán, N. (2019). Assessing variability in carbon footprint throughout the food supply chain: a case study of Valencian oranges. The International Journal of Life Cycle Assessment, 1-18.
Response 2: We agree with Reviewer 1 that there is significant variability in environmental impacts based on geographic area, management practices, and other factors. We now acknowledge this in the text, referencing the recommended papers. We read the recommended papers to inform our revision process as advised. As part of our revision process, we sought additional data to include in our analysis and, where possible, calculated mean and standard deviation across values collected for individual foods. Unfortunately, data limitations prevented us from including these for all foods included in the analysis, as some foods had only one or two values available. We noted which foods had this limitation and discussed the variability and uncertainty associated with LCA of foods. We added the standard deviation as error bars for Figure 3. We did not include this in Figure 4 due to it representing a ratio with PRO, but we include standard deviation for the GWP per serving in our Appendix or supplemental material.
Point 3: As to the serving, I agree with the authors that it reflects the amount of food consumed nowadays in the USA, but as the authors also state (line 213) they are subject to change, therefore they are also variable. In addition, serving sizes do not always agree with nutritional recommendations. Along these lines, and taking into account that according to the last USDA recommendations the, food pyramid has become a dish, it would be interesting to show some examples comparing two (or more) dishes, e.g. vegetarian vs. omnivorous.
Response 3: We appreciate the suggestion by Reviewer 1 to provide examples comparing dishes, but we believe that this would not be in line with the purpose of the paper, which is to demonstrate the importance of consideration of a specific macronutrient (protein) in the context of LCA examining individual foods. Previous studies have included consideration of meals and dietary preferences, but without our consideration of DIAAS in evaluating protein content. Ximena Schmidt Rivera et al. 2014 explored comparisons of meals in “Life cycle environmental impacts of convenience food: comparison of ready and home-made meals” and L. Baroni et al. 2006 explored comparisons of dietary patterns in “Evaluating the environmental impact of various dietary patterns combined with different food production systems”. However, our study does not focus on consumer use phase environmental impacts, but rather estimates amounts of foods consumed using RACCs and examines the associated environmental burdens of the foods themselves. We understand that some of our sources include cradle to grave system boundaries and are therefore and exception to this, but that does not change our focus. However, if a reader wished to consider the environmental impacts of a smaller or larger amount of food or certain combination of foods, it would be possible to use data in our proposed Appendix/supplemental material to calculate that for themselves. Evaluation of potential example dishes to match USDA recommendations or any other guidance would also complicate the use of the GWPRO ratio, as this is specifically to address protein quality and quantity and is therefore most appropriate for use in the context of foods commonly consumed for protein, not other foods. We note this limitation and propose the inclusion of other macronutrients in future work, but do not have an appropriate methodology for their inclusion at this time.
Point 4: In the conclusions, I recommend to add some comments about the above issues: variability of the carbon footprint and the serving size.
Response 4: We updated the conclusion to comment on the issues of variability of carbon footprints and serving sizes against actual consumption patterns as requested.
Point 5: Minor changes: Line 129. Please, include the reference of the data sources
Response 5: We updated to include references for the specific data sources used for identifying commonly consumed protein sources. We draw from two primary data sources (USDA ERS FADS) and Stefan Pasiakos et al.’s 2015 article, “Sources and Amounts of Animal, Dairy, and Plant Protein Intake of US Adults in 2007–2010” that examined data from the National health and Nutrition Examination Survey to identify the most common available protein sources and the most consumed protein sources across animal, dairy, and plant proteins, respectively.

Reviewer 2 Report
The main aim of the paper is to propose a methodology to take into consideration protein quality and quantity using the digestible indispensable amino acid score (DIAAS) when making comparisons using LCA results. This assessment has the potential to shift consumer perceptions regarding what foods are good protein sources and less environmental affecting. Since from a technical point of view LCA requires a functional unit and functional units for LCAs of food are difficult to define, the Authors consider worthwhile that, for high protein foods, a protein-based functional unit is logical, also if it is insufficient to represent the actual bioavailability of amino acids, as they acknowledge.
In order to integrate multiple dimensions of sustainability in one assessment by expanding and refining the current methodology, they propose to use reference food amounts customarily consumed (RACCs) to account for variation in amounts of food typically consumed and differences in protein quality and quantity.
To rank the considered alternatives (different foods), the Authors calculate an environmental protein efficiency (EPE) ratio, as shown in Formula 2, dividing the weighted protein score by the corresponding global warming potential (GWP), providing therefoe protein compared to the environmental impact in terms of greenhouse gas emissions. The results of the study are interesting and particularly useful for consumers, “but they may not be as readily applicable to large-scale estimates as strictly uniform weight-based comparisons normally are”, as correctly remarked by the same Authors.
My main methodological concern is putting GWP as denominator of the EPE Formula. Actually in this way the EPE values are very sensitive to small values of GWP. This effect could be easily observed comparing the results of Figures 3 and 4, where the ranking are of course quite different, and reverse especially with respect to food with very low GWP.
Could you propose different ways to aggregate these kind of values?
In my opinion, integrating in future researches into this kind of LCA study also economic considerations could be very interisting and fruitful.
Author Response
Response to Reviewer 2 Comments
Point 1: My main methodological concern is putting GWP as denominator of the EPE Formula. Actually in this way the EPE values are very sensitive to small values of GWP. This effect could be easily observed comparing the results of Figures 3 and 4, where the ranking are of course quite different, and reverse especially with respect to food with very low GWP.
Could you propose different ways to aggregate these kind of values?
Response 1: We reversed the ratio to place GWP in the numerator to address this concern. This also allows easier comparison between Figures 3 and 4 rankings, now that larger numbers in both cases are less desirable. The new ratio is called GWPRO and still captures the value of protein in making comparisons across the foods considered. We also provide additional explanation regarding the differences between Figures 3 and 4. We believe that the differences between Figures 3 and 4 are essential to the points raised in the paper – namely that incorporating consideration of protein content and quality as well as serving size together with global warming potential will result in changes in which foods are considered best and worst for their environmental impacts.
Point 2: In my opinion, integrating in future researches into this kind of LCA study also economic considerations could be very interisting and fruitful.
Response 2: We agree that integration of economic considerations would be interesting and useful in future research.

Round 2
Reviewer 1 Report
I have carefully read the reviewed paper. The authors have properly addressed the comments from the reviewers and I thus think it can be accepted for publication as it is now.